# How Environmental Factors Affect Forest Fire Occurrence in Yunnan Forest Region

Zheng Zhu [1], Xiaofan Deng [1], Fan Zhao [1,*], Shiyou Li [2] and Leiguang Wang [1]

1. College of Big Data and Intelligent Engineering, Southwest Forestry University, Kunming 650224, China
2. College of Civil Engineering, Southwest Forestry University, Kunming 650224, China
* Correspondence: fzhao@swfu.edu.cn

**Abstract:** Forest fire is an ecosystem regulating factor and affects the stability, renewal, and succession of forest ecosystems. However, uncontrolled forest fires can be harmful to the forest ecosystem and to the public at large. Although Yunnan, China is regarded as a global hotspot for forest fires, a general lack of understanding prevails there regarding the mechanisms and interactions that cause forest fires. A logistic regression model based on fire points in Yunnan detected by satellite in 2005–2019 was used to estimate how environmental factors in local areas affect forest fire events. The results show that meteorology is the dominant cause of the frequent forest fires in the area. Other factors of secondary importance are the daily minimum relative humidity and the daily maximum temperature. When using the logistic regression model based on the data of fire points in Yunnan over the period 2005–2019, the key threshold for the daily minimum relative humidity is 28.07% ± 11.85% and the daily maximum temperature is 21.23 ± 11.15 °C for a forest fire probability of 50%. In annual and monthly dynamic trends, the daily minimum relative humidity also plays a dominant role in which combustible substance load remains relatively stable from January to March, and the impact on forest fire becomes greater in April, May, and June, which plays a secondary role compared with the interannual climate. The maximum daily temperature ranks third in importance for forest fires. At the county level, minimum relative humidity and maximum temperature are the top two factors influencing forest fires, respectively. Meanwhile, the differences in forest fire points between counties correspond to the pathways of the two monsoons. This study applies quantitative expressions to reveal the important environmental factors and mechanisms that cause forest fires. The results provide a reference for monitoring and predicting forest fires.

**Keywords:** forest fire; logistic regression model; environmental factors; key threshold; quantitative expressions

## 1. Introduction

Forest fires are a key regulator of the ecological environment and an important disturbing factor in local forest landscapes [1–3]. Owing to climate change, the ecological benefits of forest fires and the main controlling factors are now of significant interest. Over the past decades, the effects of climate conditions, combustible substances, and human activities that cause forest fires have been investigated worldwide [4–7], with the focus being on the global scale of forest fire hotspots, including the northern forests of North America and northern Eurasia [8–12], tropical forest and savanna [13–17], and Mediterranean landscapes in Europe and Australia [18–20]. However, relatively few studies have considered forest fire in Southwestern China, despite this area having a high rate of forest fires. As a result, the mechanisms and interactions leading to forest fires in this region are poorly understood.

Remote-sensing multi-source data [21], combined remote sensing and ecosystem modeling [22], provincial statistical data [23], and county-level statistical data [24] show that Southwest China is a region with a high prevalence of forest fires. Yunnan is a province in Southwest China with frequent forest fires, where the main flammable species include

the widely distributed Pinus yunnanensis forest and mixed forests of green broad-leaved trees [25]. In Yunnan, the monsoon is a key factor influencing forest fires. The humid summer and autumn climate is a major feature of monsoons from the western Pacific Ocean and the Indian Ocean [26]. In winter and spring, the monsoon recedes and is replaced by warm, dry southwesterly winds [27]. Both ground monitoring and remote sensing indicate that this is the season of frequent forest fires in Yunnan [28]. Human activity is another major cause of frequent forest fires in Yunnan. In fact, slash-and-burn agriculture has been practiced in Yunnan for a long time [29,30], and many Chinese cultural traditions (e.g., burning joss paper, burning incense, and setting off firecrackers) are potential sources of fire.

The environmental variables that affect forest fire occurrence in Yunnan have rarely been investigated, so limited information is available about the causes and effects of forest fires in Yunnan. By using a remote-sensing method, Li et al. [31] determined that the distance of residences and roads from the forest zone is a key factor in the occurrence of forest fires. Xiao et al. [32] investigated lake sediments in Yunnan and found evidence of forest fire events over the last 20,000 years. They pointed out that climate conditions, combustible substances, and human activities played a crucial role in the occurrence of forest fires. Based on three years of county-level forest fire records from various provinces of China, Ying et al. [24] concluded that the presence of combustible substances was the main factor influencing the occurrence of forest fires in Yunnan. However, these studies did not consider daily meteorological data and combustible vegetation of local areas, nor did they consider multiple natural and social factors, and their related studies did not explore the mechanisms by which spatial and temporal variation in the occurrence of forest fires is associated with certain critical environmental changes.

The increasingly significant characteristics of global climate change have brought great challenges to the accurate prediction of forest fires. Therefore, it is necessary to analyze the main influencing factors in forest fire occurrence based on the background characteristics of the study area and then revise or reconstruct the original forest fire prediction model on this basis to overcome this challenge. This study considers fire points in Yunnan monitored by satellite from 2005 to 2019 and the environmental conditions at the outset of forest fires, including daily weather data, combustible substance type, terrain, and social factors, to thoroughly investigate the occurrence of forest fires in Yunnan. We also derive quantitative expressions to understand important environmental factors that affect the occurrence of forest fires and the mechanisms that cause forest fires.

Based on the above reasons, this study will consider two points to fully explore forest fires in Yunnan: First, we comprehensively consider various environmental conditions when the forest fire occurs and use quantitative expressions to reveal the important environmental factors of forest fire occurrence. The main consideration is the threshold division of environmental factors; the second is to find out the main environmental factor variables by exploring the temporal and spatial trend of forest fires so as to determine the main environmental variables of forest fires in Yunnan. The study is divided into the following sections: (1) key environmental factors and their thresholds, (2) interannual trends of forest fires, (3) intra-annual trends of forest fires, and (4) trends of spatially heterogeneous forest fires. The results of the study provide suggestions for fire prevention.

## 2. Materials and Methods

### 2.1. Study Area

Yunnan is a province in Southwest China and the provincial capital is Kunming. Its administrative area includes eight prefecture-level cities and eight autonomous prefectures, with a total of 129 county-level divisions (Figure 1). Its geographical scope is 97°31′ E–106°11′ E, 21°8′ N–29°15′ N, and the overall terrain is generally high in the northwest and low in the southeast, with altitudes ranging from 76.4 m in the southeast to 6740 m in the northwest, dominated by plateau and mountainous areas, with typical vertical geo-

graphic features and rich species diversity, making it one of the top 25 biodiversity hotspots in the world [33,34].

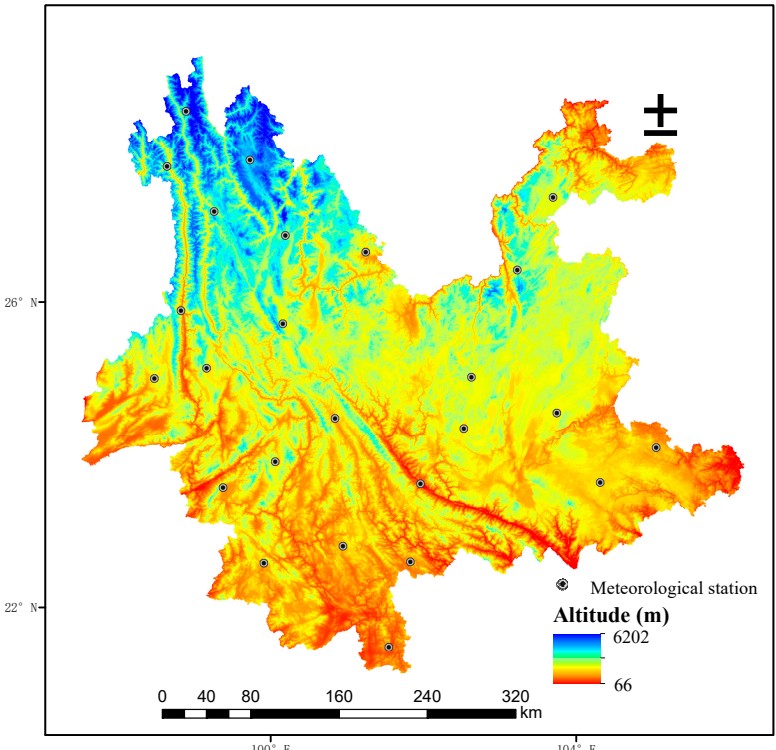

**Figure 1.** Geographical location of Yunnan and locations of major meteorological stations in Yunnan.

Yunnan has a subtropical and tropical monsoon climate, influenced by monsoons from the Western Pacific Ocean and the Indian Ocean, with distinct wet and dry seasons. The dry season lasts from November to April, which is the high-frequency period of forest fires. Additionally, Yunnan has abundant forest resources; the National Bureau of Statistics of China reports that, as of 2017, Yunnan has 19.1419 million $hm^2$ of forest area, a forest coverage rate of 50.03%, and a forest volume of 169.30919 million $m^3$. The main tree species in the Yunnan Forest Zone are *Pinus yunnanensis*, *Pinus armandii*, *Abies ernestii*, and *Quercus* L.

According to the records of the Yunnan Provincial Fire Brigade, Yunnan had the most forest fire events in the country [35]. For this study, we downloaded the entire MODIS Collection 6.1 thermal anomaly dataset covering all of Yunnan from 2005 to 2019, which was obtained from the NASA Fire Information Resource Management System (NASA FIRMS) (https://earthdata.nasa.gov (accessed on 13 September2021)). The coordinates with 95% confidence level or higher were selected because the hotspots collected by satellites were not necessarily the actual fire points. A total of 8542 fire points were further confirmed by the actual ground fire points. Upon considering the long duration of large-scale wildfire, which was recorded by satellites, 4649 fire points (Figure 1) were finally confirmed by comparing their longitude and latitude range and time span and identifying points with a time span of no more than 24 h and a distance of no more than 0.01 in longitude or latitude coordinates as the same wildfire.

### 2.2. Environmental Factors

The terrain data were obtained from SRTM 90 m digital elevation model (DEM) digital altitude database (https://srtm.csi.cgiar.org/ (accessed on 12 September 2021)), which was mainly mapped by NASA and the National Mapping Agency of the US Department of Defense through the Shuttle radar terrain [36]. The DEM image of Yunnan was obtained by cropping the data in ArcGIS 10.7. The fire points were marked into DEM images according to longitude and latitude, and the corresponding altitude, slope, and aspect of each point

were extracted. The slope aspect was divided into NS-aspect and EW-aspect. The NS-aspect and EW-aspect covered a range of 0° to 180°, and the direction was from north to south and from east to west. The meteorological data used were maximum temperature ($T_{max}$), precipitation (Preci), minimum relative humidity ($RH_{min}$), maximum wind speed (WSmax), and sunshine duration (SSD), which were obtained from the website of the China National Meteorological Information Center (http://data.cma.cn/en (accessed on 8 June 2020)), which has 25 basic stations (Figure 1). Meanwhile, considering the combustion properties of local combustible substances, the annual cumulative gross primary productivity (GPP) of each fire point [37] was calculated and extracted, starting from the year corresponding to the date of the fire points. These data were obtained from NASA's MO/YD17A2 product (https://search.earthdata.nasa.gov/ (accessed on 15 September 2021)).

Considering the difference in combustible substance types, 30 m global ground cover type data (https://www.webmap.cn/ (accessed on 15 September 2021)) was used to extract the combustible substance type corresponding to each fire point. The intensity of human activity was estimated by using the distances from fire points to residences and road networks, so the 1:250,000 National Basic Geographic Information Database (https://www.webmap.cn/ (accessed on 15 September 2021)) was imported into Arcgis 10.7 to extract the distances to residences and roads. Table 1 lists the 12 factors used in this study.

**Table 1.** Data source.

| Variable Type | Variable Factor | Data Sources | Unit |
|---|---|---|---|
| Meteorological | Precipitation | China National Meteorological Information Center | mm |
| | Maximum temperature | | °C |
| | Minimum relative humidity | | Relative humidity (RH%) |
| | Maximum wind speed | | m/s |
| | Sunshine duration | | h |
| Terrain | Altitude | Digital elevation database | m |
| | Slope | | ° |
| | NS-Aspect | | ° |
| | EW-Aspect | | ° |
| Combustible substance | GPP | NASA | kg C |
| Human activity | Distance from residence | 1:250,000 National Basic Geographic Information Database | m |
| | Distance from road | | m |

*2.3. Research Method*

Estimating the probability of wildfire from only existing data requires random non-fire points and fire points for the sample data. As per previous studies [37–39], the ratio of fire points to non-fire points was adjusted to 1:1.5. To do this, we added 6974 random non-fire points randomly generated in ArcGIS 10.7, and the dates of these non-fire points were randomly assigned using Excel to ensure that all points were unique in terms of time and location. The environmental factor was then extracted in the same way as the fire point samples. The logistic regression (LR) model reveals the relationship between wildfire ignition and environmental factor, and it has been widely applied to study wildfires [40,41]. Although researchers have used various methods to study forest fire probability ignition [42,43], the LR model has been used more often in empirical studies because of its simplicity and its reliable results. Most importantly, the LR model can be used to establish, based on the data, mathematical equations for the forest fire probabilities and other environmental variables

and then to determine the threshold values of the corresponding variables for 50% forest fire probability and the partial derivatives when the forest fire probability is 50%. Therefore, an LR model with 12 environmental variables was established to clarify the relationship between each environmental variable and forest fire probability [44]:

$$\ln(\frac{P}{1-P}) = \beta_0 + \sum_{j=1}^{n} \beta_j x_j \tag{1}$$

where $P$ is the forest fire probability and $1 - P$ is the probability of no forest fire, $\beta_j$ is the regression coefficient of environment variable $j$, $x_j$ is environment variable $j$, and $n$ is the total number of environment variables.

The receiver–operator characteristic curve is often used to evaluate the accuracy of the classification model and to calculate the optimized threshold [45]. It uses a horizontal coordinate of specificity (a positive class misclassified as a negative class) and a vertical coordinate of sensitivity (a positive class judged as a positive class), and the area under the curve (AUC) of the receiver–operator characteristic curve serves to evaluate the accuracy of the model. The range of AUC is 0–1, and the model has good predictive ability when AUC > 0.8 [46–48]. Given the possible collinearity problem between environmental variables, which decreases the accuracy of the model, we apply a multiple collinearity diagnosis to eliminate the relevant variables. The random forest algorithm serves to rank the importance of the environmental variables filtered by the model. The importance of each explained variable is expressed in terms of the decrease in the AUC when the variables are randomized 500 times [49,50].

The mathematical interpretation of the LR model leads to two important features of the derived mathematical equation, which should be further analyzed. The first important feature is the marginal threshold of each environmental variable when the forest fire probability is 50%, which can be calculated in the reverse direction by using mathematical equation. When the forest fire probability is 50%, the marginal value of each environmental variable can be calculated by using the practical values of all other variables in the model. This is the marginal threshold of the given environmental variable. The second important feature is to determine the partial derivative of the curve when the forest fire probability is 50%; this partial derivative reflects how the environmental variables affect the 50% forest fire probability. The larger the partial derivative, the greater the influence of environmental factors on the occurrence of forest fires. Finally, all statistical analyses were done using SPSS.26 (IBM, Chicago, IL, USA) and Rx 64 4.1.2 (Auckland University, Auckland, New Zealand).

## 3. Results

In this study, the study area and data samples were first determined, the LR model was further constructed to obtain the environmental factors with no significant correlation, and the corresponding marginal thresholds and corresponding partial derivatives were calculated. Finally, the relative importance of each factor was displayed by the random forest method. Proportionally, the following conclusions are explored from the total data samples of the whole year, the data samples of annual and monthly dynamics, and the data samples of the county scale.

### 3.1. Spatiotemporal Distribution and Causes of Fire Points

A total of 4649 forest fire events were detected in 129 counties of Yunnan Province during the period 2005–2019, with great spatial variability. Specifically, fire points were concentrated in Northwest Yunnan, Southwest Yunnan, and Southeast Yunnan (Figure 2a) On average, 310 fire events occurred per year, although 842 fires occurred in 2010, accounting for more than 18% of all forest fires (Figure 2c). Forest fires mainly occurred between December and June, with more than 99% of forest fires occurring in the first half of the year

(Figure 2d). The fire season in Yunnan was from 1 December to 15 June, and the number of forest fires in March accounted for more than 26% of the annual number of forest fires.

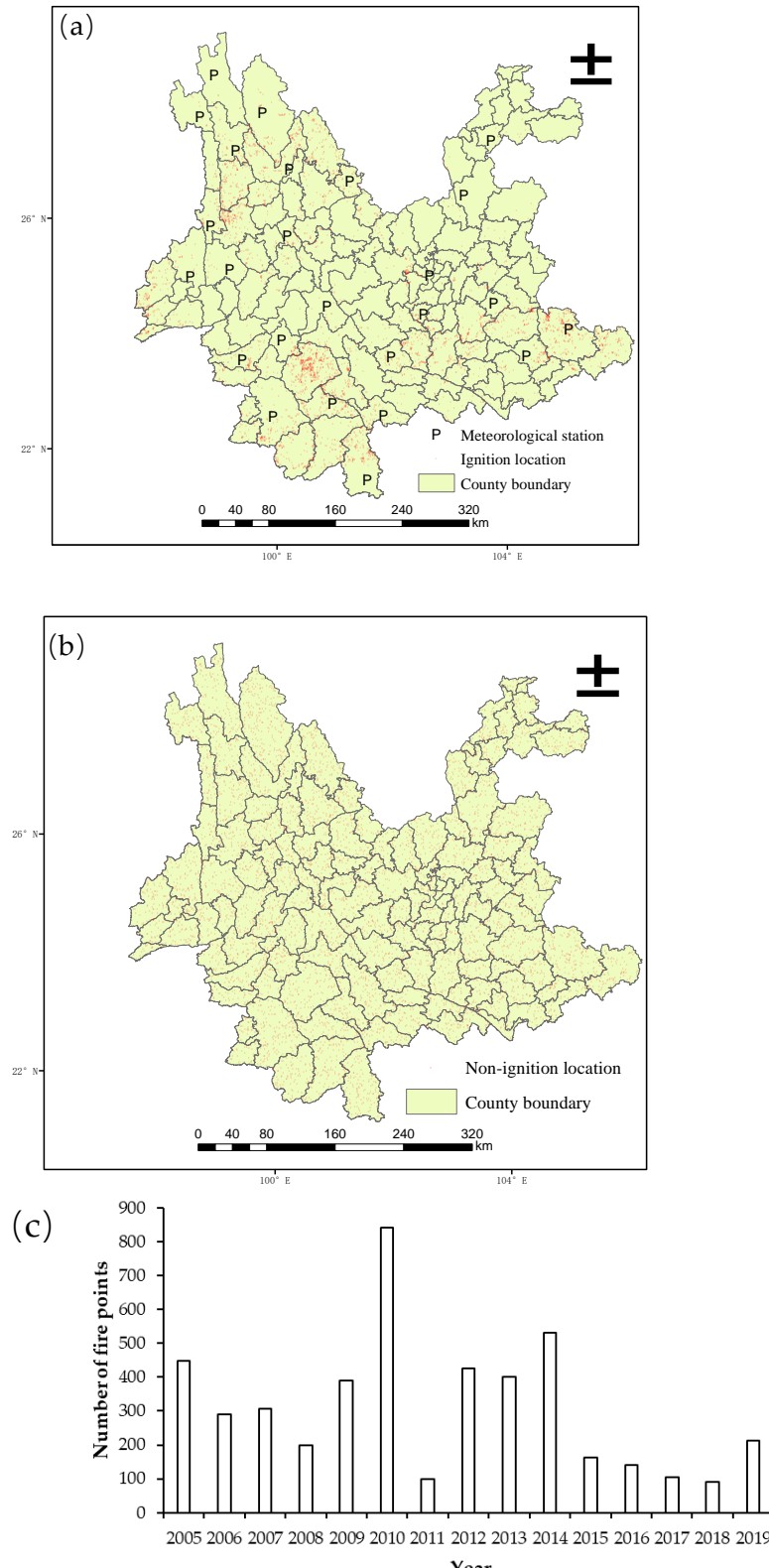

**Figure 2.** *Cont.*

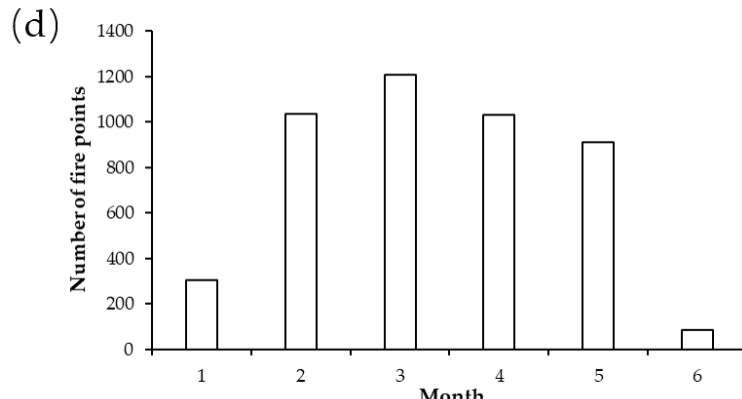

**Figure 2.** (**a**) Distribution of fire points in forest as detected by satellite in Yunnan, China, 2005–2019. (**b**) Random non-fire locations generated by ArcGIS. (**c**) Number of fire points for each year, 2005–2019. (**d**) Number of fire points for each month, January–June, averaged over the years 2005–2019.

Multicollinearity and significance tests ($p < 0.05$) were performed on the full samples, producing a total of 10 independently correlated variables (see Table 2). According to the results of the LR model and random forest ranking, it is shown that the AUC of the LR model constructed from the full sample was 90.82%, and the most important environmental factor was $RH_{min}$ (Figure 3), which correlates negatively with forest fire occurrence and has a key threshold of 28.07% ± 11.85% (Table 2) at 50% forest fire probability. The correlation partial derivative indicates that a 1% decrease in $RH_{min}$ increases the marginal forest fire probability by 3.04%. The second major factor is $T_{max}$ (Figure 3), followed by GPP, with the relative importance between the two being less than 0.1%. For these variables, the key threshold for 50% forest fire probability is 21.23 ± 11.15 °C and $2.58 \times 10^3 \pm 1.42 \times 10^3$ kg K, respectively. The associated rates of change of the partial derivative are 1.74% °C$^{-1}$ and 0.03% (kg K)$^{-1}$, which both correlate positively with forest fire probability. This section may be divided by subheadings. It should provide a concise and precise description of the experimental results, their interpretation, as well as the experimental conclusions that can be drawn.

**Table 2.** Fitting eigenvalues for full data logistic regression model.

| Variables | Regression Coefficients | Threshold (±Standard Deviation) of 50% IgnitionProbability | Partial Derivative |
|---|---|---|---|
| Precipitation | $-1.52 \times 10^{-1}$ | 8.16 (±6.46) | −3.79 |
| Minimum relative humidity | $-1.22 \times 10^{-1}$ | 28.07 (±11.85) | −3.05 |
| Sunshine duration | $4.3 \times 10^{-2}$ | 7.32 (±3.43) | 1.07 |
| Maximum temperature | $6.8 \times 10^{-2}$ | 21.23 (±11.15) | 1.70 |
| Maximum wind speed | $1.34 \times 10^{-1}$ | 5.32 (±2.19) | 3.35 |
| Distance from residence | $2.26 \times 10^{-4}$ | $1.33 \times 10^3$ ($\pm 1.07 \times 10^3$) | $5.00 \times 10^{-3}$ |
| Distance from road | $9.57 \times 10^{-5}$ | $2.39 \times 10^3$ ($\pm 1.79 \times 10^3$) | $2.50 \times 10^{-3}$ |
| Slope | $9.63 \times 10^{-3}$ | 46.32 (±24.52) | $2.50 \times 10^{-1}$ |
| Altitude | $-6.49 \times 10^{-5}$ | $2.45 \times 10^3$ ($1.12 \times 10^3$) | $1.50 \times 10^{-3}$ |
| GPP | $1.28 \times 10^{-3}$ | $2.59 \times 10^3$ ($1.22 \times 10^3$) | $3.25 \times 10^{-2}$ |
| Constant | −2.23 | - | - |

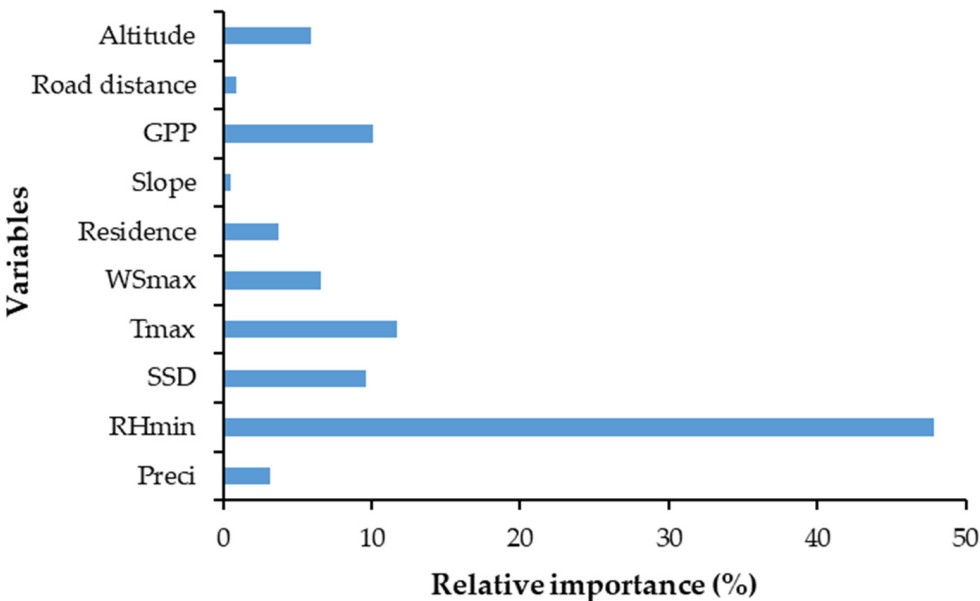

**Figure 3.** Ranking of relative importance of environmental variables.

### 3.2. The Annual and Monthly Dynamic Trends of Environmental Factors

In the period 2005–2019, the AUC of the LR model exceeds 90% in all years, and $RH_{min}$ is the most important environmental factor affecting the forest fire probability in the model in all years (Figure 4a). GPP and $T_{max}$ are the second and the third most important environmental factors, respectively. In most years, the relative importance of $RH_{min}$ exceeds 50% (Figure 3), and the relative importance of $RH_{min}$ varies significantly between years, with 2010 having the largest relative importance, with 72% of the total, and 2018 having the lowest at 33.46% of the total. The Spearman correlation coefficient for $RH_{min}$ and GPP indicates that no significant relationship exists between the two ($p = 0.416$), $RH_{min}$ correlates negatively with $T_{max}$ ($r = -0.67$, $p = 0.007$), and the corresponding values of $RH_{min}$ threshold and forest fire for 50% of the train were not significant, and their inter-annual variation is not significant (Figure 4b,c).

Between January and June, over 98% of forest fires occurred in the first half of the year (Figure 2c), whereas the results for the other months are not statistically significant because of the small amount of data. Therefore, we use herein the LR model from January to June to study the monthly dynamics of forest fires. $RH_{min}$ is the most important environmental factor from January to May (Figure 5a), during which time the AUC of the related model exceeds 90% (except for April, which is 85%). GPP is the second-most important among the relevant models from April to May (Figure 5a), and it becomes the most important environmental factor in June, with an AUC of 93.5% for this model. No significant correlation exists between these two factors ($p = 0.208$), and the variation of $RH_{min}$ and the threshold of 50% forest fire probability is more significant, reaching its maximum threshold in June. The GPP threshold at 50% forest fire probability is approximately $2.7 \times 10^3$ (kg K) (Figure 5b) and does not change significantly from January to June. The partial derivative of $RH_{min}$ and GPP at 50% forest fire probability also does not change significantly (Figure 5b,c, respectively).

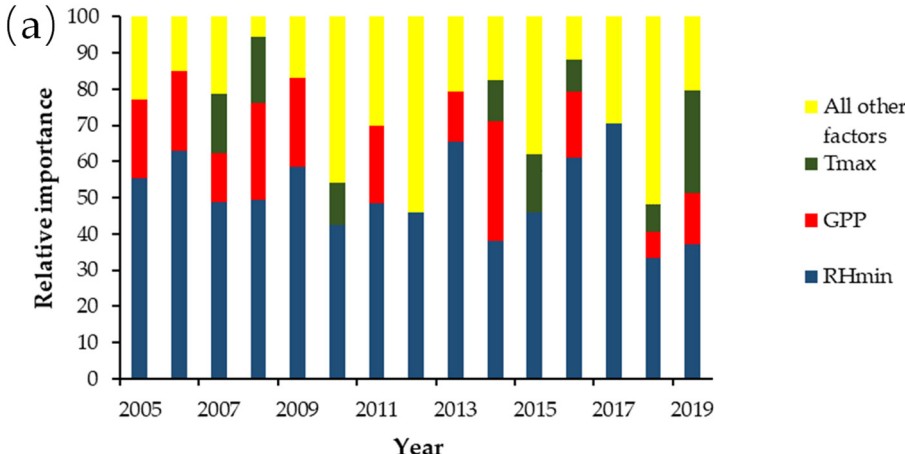

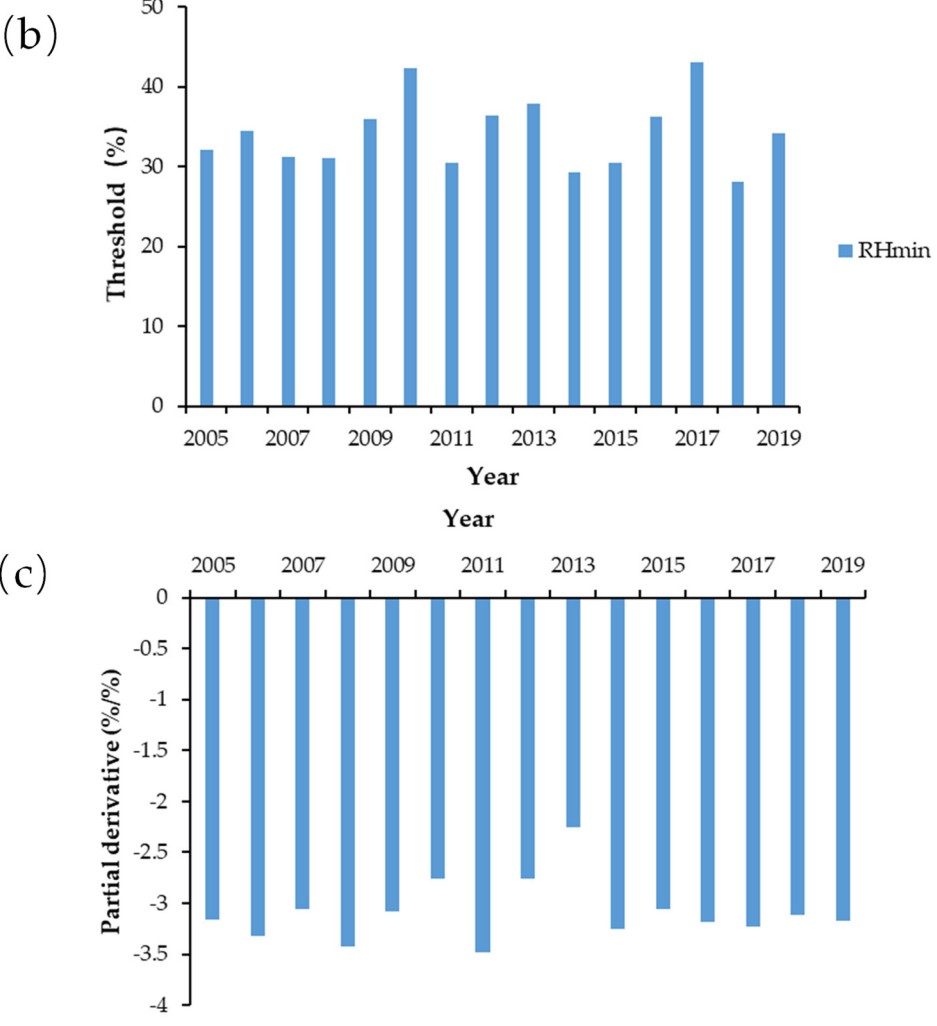

**Figure 4.** (**a**) Ranking of relative importance of environmental variables for each year. (**b**) Interannual thresholds for the most important environmental variables for each year. (**c**) Interannual partial derivatives of the most important environmental variables for each year.

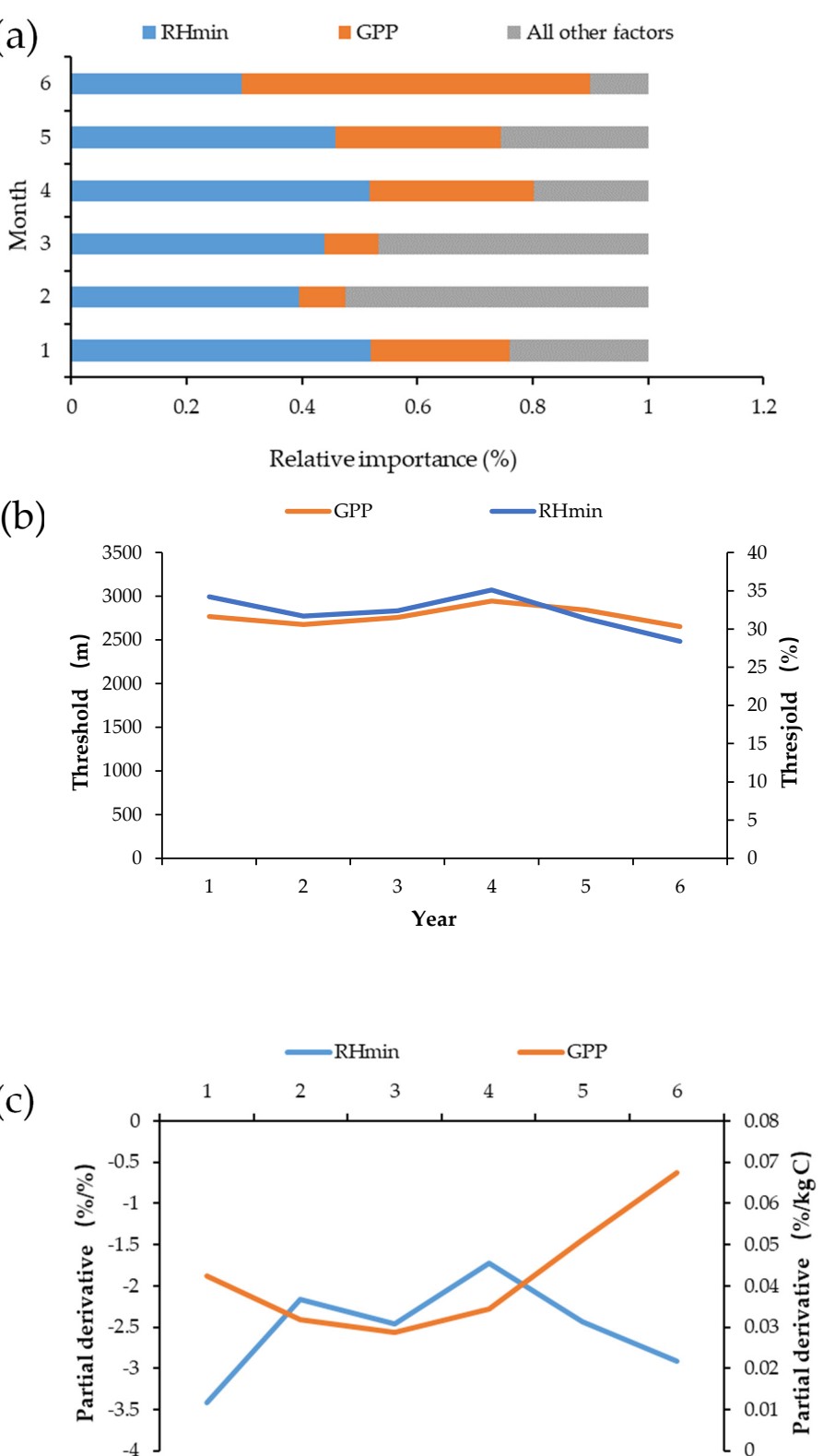

**Figure 5.** (**a**) Ranking of relative importance of environmental variables by month in the first half of the year. (**b**) Thresholds of RH_min and GPP for each month in the first half of the year. (**c**) Partial derivatives of RH_min and GPP in each month of the first half of the year.

### 3.3. Spatial Heterogeneity of Environmental Factors

This study establishes 94 county-level LR models; 23 other counties have too few fire points to be statistically significant. Thus, the spatial heterogeneity of the environmental factors was analyzed by using the 94 abovementioned county-level LR models. The AUCs of all these county-level LR models exceed 90%, and the top three environmental factors for the county-level models are $RH_{min}$, $T_{max}$, and altitude (Figure 6a). $RH_{min}$ plays a dominant role in most county-level models, with an overall bias toward Northwest Yunnan, Southwest Yunnan, and Southeast Yunnan. $T_{max}$ is the dominant factor in Central Yunnan and in a few regions in Northwest Yunnan and Northeast Yunnan. Altitude is the dominant factor in Central Yunnan and in a few counties in Northeast and Southeast Yunnan. A correlation analysis of 63 county-level models reveals a significant negative correlation between $RH_{min}$ and the relative importance coefficient of $T_{max}$ ($r = -0.61$, $p = 0.003$), whereas no significant relationship appears between the other two pairs of correlation factors. The correlation coefficients of $RH_{min}$ and altitude are not significant in 54 counties ($p = 0.372$) and $T_{max}$ and altitude are not significantly correlated in 40 counties ($p = 0.235$; Figure 6b–d).

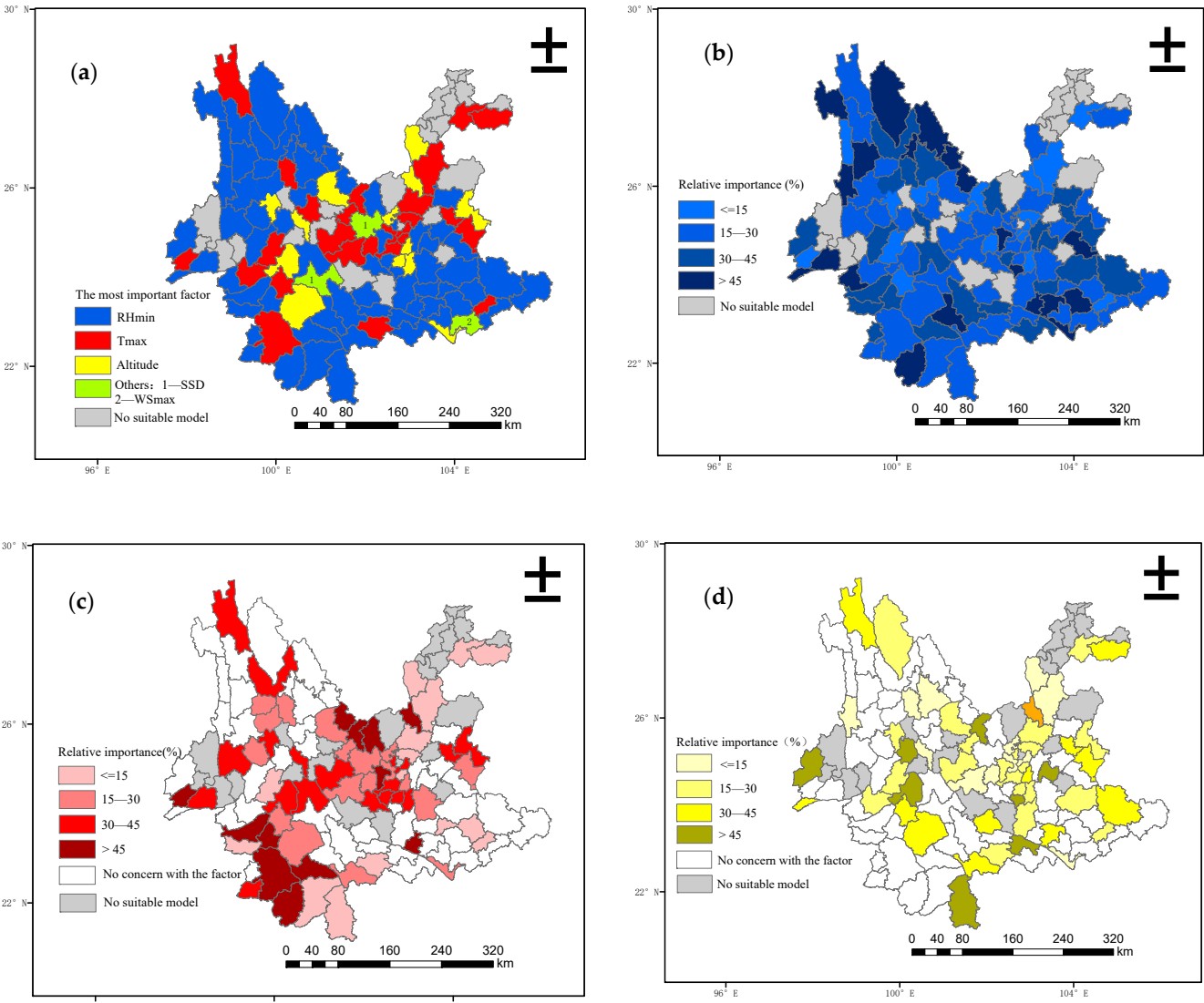

**Figure 6.** (**a**) Most important environmental variables in different counties. (**b**) Ranking of relative importance of $RH_{min}$ in each county. (**c**) Ranking of relative importance of $T_{max}$ in each county. (**d**) Ranking of relative importance of altitude in each county.

At 50% forest fire probability, the threshold of $RH_{min}$ trends downward from Northwest Yunnan, through Southwest Yunnan and Southeast Yunnan, and finally to Central Yunnan (Figure 7a), where the partial derivative $RH_{min} < -8$ mainly occurs in Northwest Yunnan, Southeast Yunnan, Southwest Yunnan, and in the areas around the Lancang River and the Nujiang River, which indicates that for every 1% decrease in $RH_{min}$, the marginal forest fire probability of 50% in this region increases by at least 8% (Figure 7b). The high $T_{max}$ threshold mainly occurs in Central Yunnan, Northwest Yunnan, and Southwest Yunnan, decreasing in these three regions upon approaching the surrounding area. The highly correlated partial derivative is concentrated in Central Yunnan and Southwest Yunnan, and its marginal variation is much less than that of $RH_{min}$ (Figure 7c,d). Compared with $RH_{min}$ and $T_{max}$, the altitude threshold for 50% forest fire probability is relatively large, and the degree of variation of the threshold and the related partial derivative reveals no obvious regularity. Overall, the variation of threshold and the related partial derivatives differ between counties (Figure 7e,f).

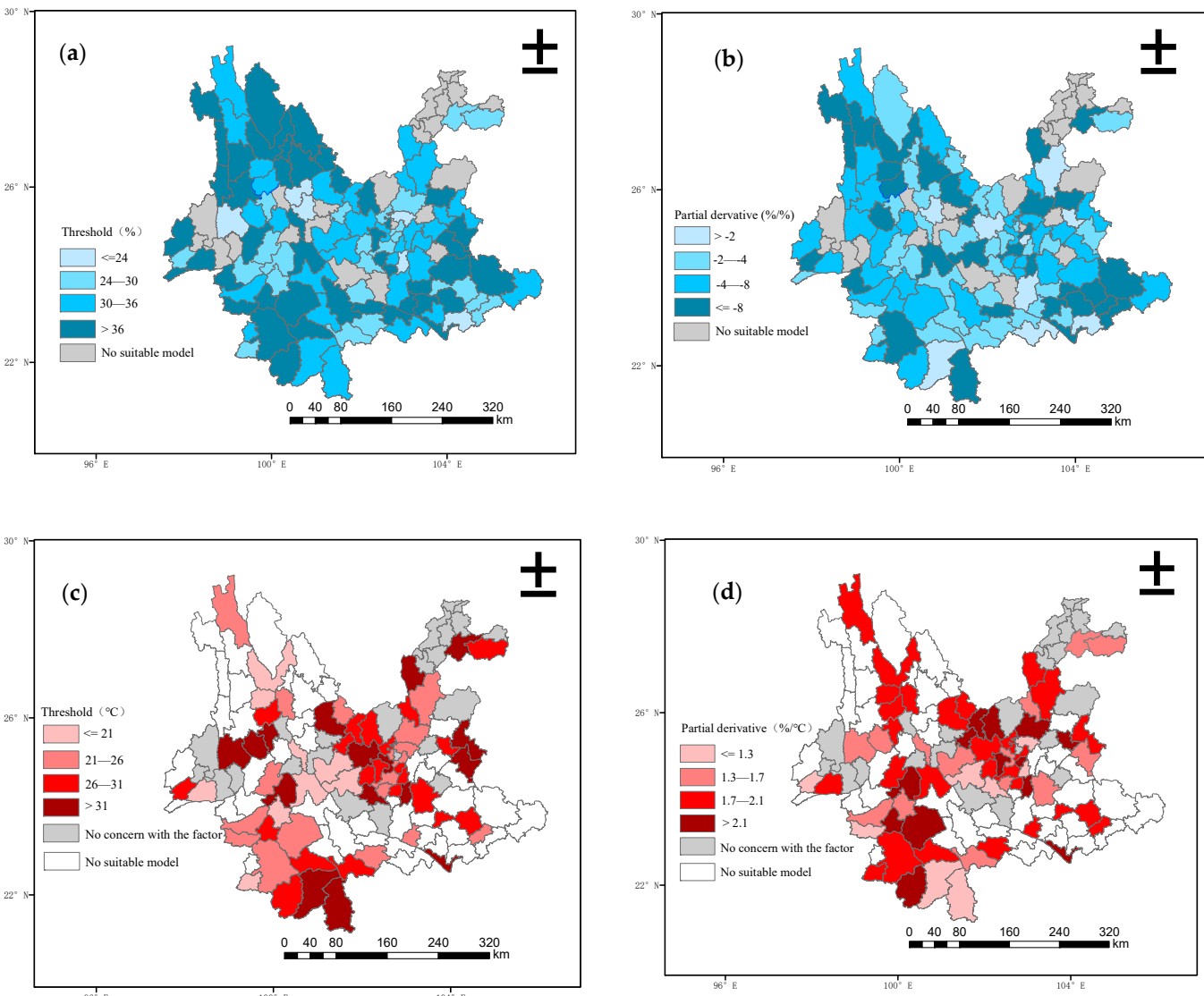

**Figure 7.** *Cont.*

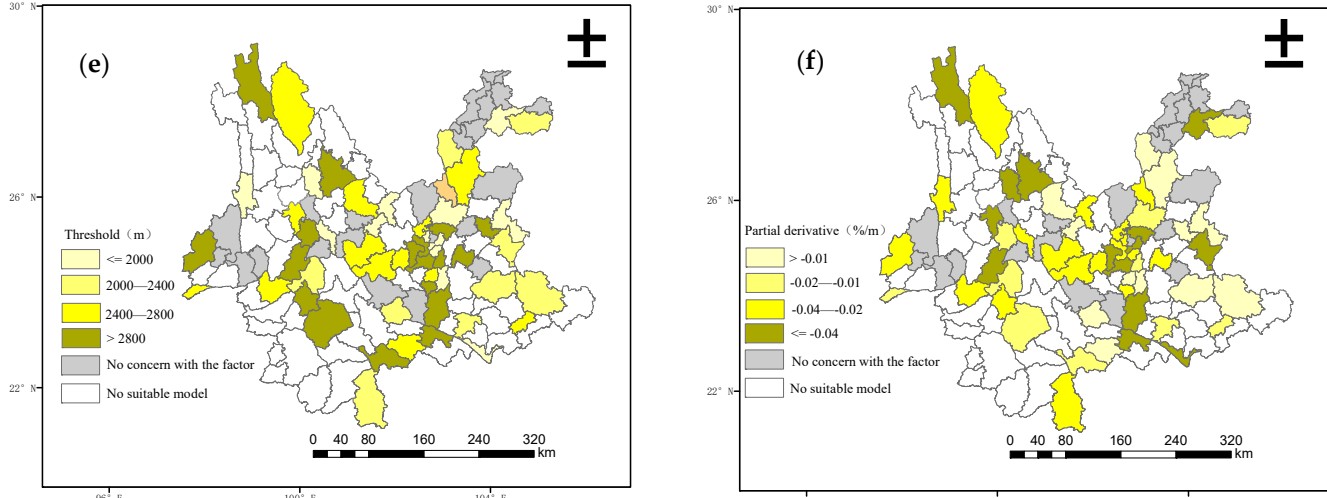

**Figure 7.** (**a**) Ranking of RH$_{min}$ thresholds by county. (**b**) Ranking of partial derivatives of RH$_{min}$ by county. (**c**) Ranking of T$_{max}$ thresholds by county. (**d**) Ranking of partial derivatives of T$_{max}$ by county. (**e**) Ranking of altitude thresholds by county. (**f**) Ranking of partial derivatives of altitude by county.

## 4. Discussion

Based on previous studies, two main driving mechanisms are responsible for igniting general forest fires [18,51]: one is drought, which generally triggers forest fires in areas with a copious and spatially continuous accumulation of combustible substance, and the other is the fire-source mechanism, which involves ignition of a combustible substance by other means, thus triggering a forest fire [39,52,53]. The latter mainly occurs in ecosystems with relatively little accumulation of combustible substance and relatively low productivity [54,55]. The climatic characteristics of Yunnan are mainly tropical and subtropical, which are more consistent with the drought-driven mechanism.

The results of this study show that RH$_{min}$ is the most important factor causing forest fire, followed by maximum temperature, whereas other factors, such as GPP and altitude, make limited contributions (Figure 3). On an interannual scale, RH$_{min}$ is also an important factor that causes forest fires throughout the year; the second most important interannual factor is GPP (Figure 4). In addition, the importance of RH$_{min}$ undergoes a significant interannual variation, which indicates that other factors, especially vegetation cover and the accumulation of combustible substance, were relatively more variable over the time interval studied [56].

According to previous studies, most of the forest fires in the study were anthropogenic [57], suggesting that most of the farmland and forest in Southern China coexist in the form of a mosaic, with this mixed landscape being particularly common in mountainous areas. Forest fires due to agricultural activities were very common [58], and agricultural activities in Yunnan were also the main cause of forest fires during this study interval, especially at the peak of the farming season (April to June). GPP becomes the second-most-important factor affecting the occurrence of forest fire in April and May and surpasses RH$_{min}$ as the most important factor in June. April to June is the growth period of most vegetation, so the combustible load increases during this period. The significantly higher summertime maximum temperature then makes forest fires easy to trigger. Traditional festivals, such as the Qingming Festival in China, which involves burning incense at graves, can then lead to numerous forest fires. In months other than June (i.e., January to May), RH$_{min}$ is the main factor affecting the occurrence of forest fire, which indicates that the combustible substance in the region was suffering from a prolonged drought, thus establishing environmental conditions ideal for forest fires.

At the county level, $RH_{min}$ and $T_{max}$ may play a spatially complementary role in determining the exact location of forest fires (Figure 6). With the influx into Yunnan of monsoons from the Western Pacific Ocean and the Indian Ocean, $RH_{min}$ increases significantly [59,60], and the relative importance of $T_{max}$ becomes more pronounced in Central and Southwestern Yunnan, where frequent droughts occur [61,62]. Meanwhile, the threshold of $RH_{min}$ decreases upon going from Northwest Yunnan to Southeast Yunnan, corresponding to the path of monsoons. Given the change in the threshold gradient, the load of vegetation and other combustible substance in this area is relatively low, therefore, drier climatic conditions are required to initiate forest fires, and the initiation of forest fire depends more on the combustibility level of the combustible substance. According to previous studies, the combustibility of different tree species varies significantly [63,64]; oil pine forests are more likely to ignite, which strongly affects the forest fire probability in the region. Altitude dominates as a fire-causing factor in a few county-level models, which might be due to human activities in these tropical forest zones, bringing fire sources and thus triggering forest fires [65]. Additionally, in alpine areas, arson also occurs to maintain grazing grassland [66].

## 5. Conclusions

In Yunnan, the occurrence of forest fires has been controlled by local environmental factors and is driven by meteorological characteristics, among which $RH_{min}$ is the dominant factor. Based on the LR model established by data of fire points in Yunnan from 2005 to 2019, the key threshold of $RH_{min}$ was 28.07% $\pm$ 11.85% at the 50% level of forest fire probability. $RH_{min}$ was the most important environmental factor affecting the occurrence of forest fires in Yunnan over the years studied. The importance of $T_{max}$ for forest fire events ranks second. The key threshold of $T_{max}$ over the period studied was 21.23 $\pm$ 11.15 °C, and the peak forest fire period in Yunnan was in spring and summer, when Yunnan has its dry season. Thus, the combustible substance was relatively dry, so $T_{max}$ easily became a natural source of forest fire. $RH_{min}$ and $T_{max}$ are spatially complementary, and the path of monsoons from the Western Pacific Ocean and the Indian Ocean corresponds to areas with the highest $T_{max}$ threshold. The results of this study allow us to use the key thresholds to determine the local early warning signs of forest fire and provide suggestions for preventing forest fires.

Based on this analysis of the results of this study, we propose several fire protection measures for Yunnan. First, in the LR model of data from Yunnan during 2005–2019, the most important environmental factor is $RH_{min}$. Therefore, forest fire monitoring and risk assessment should focus more on changes in the $RH_{min}$ threshold and daily relative humidity, especially in the Eastern and Western regions of Yunnan, where monsoons have the greatest influence. Fire-prevention services should also design the distance between regional roads and the forest zone to improve forest fire prevention measures. Controlling the sources of fire in the forest zone is also a vital step to curb fire outbreaks. According to recent statistical data, human-caused forest fires have become more frequent [21], even exceeding the number of forest fires caused by climate change [67], and the use of fires for farming and culture in spring and summer, especially in the agricultural and forest lands, should be a major concern for the authorities. To better understand the combustibility of combustible substances in different regions, combustibility experiments should be implemented on different plant species to understand how they adapt to different degrees of forest fire, and also to raise public awareness of forest fire prevention through related scientific activities. Additionally, the management strategy should be tailored to the characteristics of each region (e.g., the strong monsoons of the Western and Southeastern parts of Yunnan), and meteorological data reports should be fully exploited. For the central plateau of Yunnan and the area where the three rivers merge, forestry engineering should be adopted to reduce the combustible substance load by appropriately adjusting the forest stand structure. For a tropical forest zone and alpine grassland areas, more reasonable

management measures of the forest zone and livestock should be implemented to reduce forest fires.

**Author Contributions:** Conceptualization, Z.Z. and F.Z.; funding acquisition, F.Z.; writing—reviewing and editing, Z.Z., X.D. and F.Z.; field investigation, experiment, and data analysis, Z.Z., S.L., L.W. and F.Z.; All authors have read and agreed to the published version of the manuscript.

**Funding:** This study was financially supported by the National Natural Science Foundation of China (Grant No. 32160374) and Yunnan Fundamental Research Projects, China (Grant No. 20210AT070045).

**Institutional Review Board Statement:** Not applicable.

**Informed Consent Statement:** Not applicable.

**Data Availability Statement:** Not applicable.

**Conflicts of Interest:** The authors declare no conflict of interest.

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
