# Peer review of "How Environmental Factors Affect Forest Fire Occurrence in Yunnan Forest Region"

_forests, doi:10.3390/f13091392_

Round 1
Reviewer 1 Report
This study tried to analyze the driving factors of forest fire in Yunnan province, China using remote sensing and environmental datasets. This is an interesting study. The topic of this manuscript fits this journal. I have the following comments, which may help improve this study.
Correlation analyses between forest fire and environmental factors do not mean casual relationships or mechanisms.
Page 113-113: It is unclear how did this study confirmed those fire points using actual ground fire points. Are these fire points forest fires?
Figure 2 needs to add a graph to show non-forest fire points.
Figure 3 and other figures, it is unclear how the relative importance is calculated in this study.
Figure 5, Thresjold should be Threshold?
Reviewer 2 Report
This paper follows the scope of the Forest. The content offers input on forest fire mitigation. Small parts still need improvement:
· Several journal articles have discussed forest fire in Yunnan related to climate change, precipitation regime, the use of logistic regression (LR) in forest fire prediction, etc., so in this article, the author should convey what novelty is offered in this paper.
· Use italics for the Latin names of plant species.
· The method should be arranged more systematically following the subsections in the results section.
· The source/reference of equation 1 should be mentioned.
· Line 160-162: equation symbols should be completed.
· In Figure 1 and other maps, does the symbol for the north direction use the plus-minus sign?
· Add meteorological station points in Figure 2. a).
· The conclusions should focus on the main results and answer the research objective.
Round 2
Reviewer 1 Report
I have no further comments. Thanks.